# Basis profile curve identification to understand electrical stimulation effects in human brain networks

**Kai J. Miller**[1,2]*, **Klaus-Robert Müller**[3,4,5,6], **Dora Hermes**[2]

**1** Department of Neurological Surgery, Mayo Clinic, Rochester, Minnesota, United States of America, **2** Department of Biomedical Engineering & Physiology, Mayo Clinic, Rochester, Minnesota, United States of America, **3** Google Research, Brain Team, Berlin, Germany, **4** Machine Learning Group, Department of Computer Science, Berlin Institute of Technology, Berlin, Germany, **5** Department of Artificial Intelligence, Korea University, Seoul, Republic of Korea, **6** Max Planck Institute for Informatics, Saarbrücken, Germany

\* miller.kai@mayo.edu

**Data Availability Statement:** Code written in MATLAB to reproduce all of the steps and illustrations contained in this manuscript is freely available along with the sample dataset at https://

## Abstract

Brain networks can be explored by delivering brief pulses of electrical current in one area while measuring voltage responses in other areas. We propose a convergent paradigm to study brain dynamics, focusing on a single brain site to observe the average effect of stimulating each of many other brain sites. Viewed in this manner, visually-apparent motifs in the temporal response shape emerge from adjacent stimulation sites. This work constructs and illustrates a data-driven approach to determine characteristic spatiotemporal structure in these response shapes, summarized by a set of unique "basis profile curves" (BPCs). Each BPC may be mapped back to underlying anatomy in a natural way, quantifying projection strength from each stimulation site using simple metrics. Our technique is demonstrated for an array of implanted brain surface electrodes in a human patient. This framework enables straightforward interpretation of single-pulse brain stimulation data, and can be applied generically to explore the diverse milieu of interactions that comprise the connectome.

## Author summary

We present a new machine learning framework to probe how brain regions interact using single-pulse electrical stimulation. Unlike previous studies, this approach does not assume a form for how one brain area will respond to stimulation in another area, but rather discovers the shape of the response in time from the data. We call the set of characteristic discovered response shapes "basis profile curves" (BPCs), and show how these can be mapped back onto the brain quantitatively. An illustrative example is included from one of our human patients to characterize inputs to the parahippocampal gyrus. A code package is downloadable from https://purl.stanford.edu/rc201dv0636 so the reader may explore the technique with their own data, or study sample data provided to reproduce the illustrative case presented in the manuscript.

purl.stanford.edu/rc201dv0636, for use without restriction.

**Funding:** KJM was supported by the Van Wagenen Fellowship, the Brain Research Foundation with a Fay/Frank Seed Grant, and the Brain & Behavior Research Foundation with a NARSAD Young Investigator Grant. This work was also supported by NIH-NCATS CTSA KL2 TR002379 (KJM). DH was supported by the NIH-NIMH CRCNS R01MH122258-01. Manuscript contents are solely the responsibility of the authors and do not necessarily represent the official views of the NIH. KRM was supported in part by the Institute of Information & Communications Technology Planning & Evaluation (IITP) grant funded by the Korea Government (No. 2017-0-00451, Development of BCI based Brain and Cognitive Computing Technology for Recognizing User's Intentions using Deep Learning) and (No. 2019-0-00079, Artificial Intelligence Graduate School Program, Korea University), and by the German Ministry for Education and Research (BMBF) under Grants 01IS14013A-E, 01GQ1115, 01GQ0850, 01IS18025A, 031L0207D and 01IS18037A; the German Research Foundation (DFG) under Grant Math+, EXC 2046/1, Project ID 390685689. The funders had no role in study design, data collection and analysis, decision to publish, or preparation of the manuscript.

**Competing interests:** The authors have declared that no competing interests exist.

## Introduction

Brain networks have been explored electrophysiologically with a variety of techniques, spanning a variety of spatial scales, such as electroencephalography (EEG), magnetoencephalography (MEG), intracranial EEG (iEEG), and microelectrode local field potentials (LFPs). Efforts to infer connectivity between brain regions may search for correlated signals in response to supervised perturbation by a behavioral task, or in an unsupervised state ("resting" awake, or sleeping). Alternately, it has been shown that interactions between brain regions may be studied by applying or inducing pulses of electrical stimulation to a particular site, while measuring the electrophysiological response elsewhere [1–3]. In recent years, a sub-field of neuroscience has matured around systematic stimulation and measurement through implanted (iEEG) arrays of brain surface (electrocorticography, ECoG) or deeply-penetrating (stereoelectroencephalography, SEEG) electrodes, typically called "cortico-cortical evoked potentials" (CCEPs) or, for the special case of short pulses separated by several seconds, "single-pulse electrical stimulation" (SPES) [4–6]. The more general term, "CCEP" will be used to refer to both in this text.

For an array of $N$ total electrodes, there are a potential set of order $N^2$ CCEP interactions that may be explored (for bipolar stimulation, the exact number will depend on how neighboring electrode pairs are chosen for stimulation). We organize the approach to CCEP data into a few different network paradigms, illustrated in Fig 1. In the "all-to-all" case where one wishes to examine the full set of $N^2$ interactions (incorporating the temporal property of each response), the limited number of stimulation events possible to record in the clinical environment (where these measurements are made) does not allow for a well-defined exploration of the network. Therefore, scientists have imposed a type of constraint, or a hybridization of several constraints. One such constraint is to reduce the problem by beginning the data exploration with a pre-defined interaction based on location, and then to study the temporal dynamics within that paired framework ("hypothesis preselected"—Fig 1D) [4].

A "divergent" paradigm is commonly adopted, where the effect of stimulating a chosen site on all of the other sites is used to infer motifs of connectivity. However, the underlying cortical laminar architecture of each recipient (measured from) site is very different, and therefore the voltage timecourse of each CCEP cannot be interpreted in a common "physiological language" to distinguish different types of interactions [7]. Nonetheless, many studies have found it useful to impose a constraint by assuming a canonical form (temporal structure) and then parameterize within this assumed form (e.g. voltage at fixed delay time from stimulus). The most common of these is a negative deflection between $\sim 10$–$100$ ms (or "N1") and a later second negative deflection ("N2") to characterize which brain regions are connected [4]. That approach forces each response into a fixed interpretation where the form being fit may not actually be present. Notably, we have found that the N1/N2 response shape is not a universal phenomenon: when observed, this shape is only one of a wide variety of responses at any given brain site.

A different, "convergent", paradigm is to focus on measurement from a single site, and examine the effect of stimulating in the remainder of the array [8]. The convergent approach is, in principle, more tractable than other paradigms. When one measures from a single site and stimulates many, there is only one "physiological language" that evoked responses must be interpreted in, because the underlying anatomy is unchanged. In this context, different shaped temporal responses must imply a fundamentally different nature of interaction between brain areas.

With the convergent paradigm as our initial framework, we made the visual observation that inputs to primary motor cortex had very different shaped CCEPs from one another, but

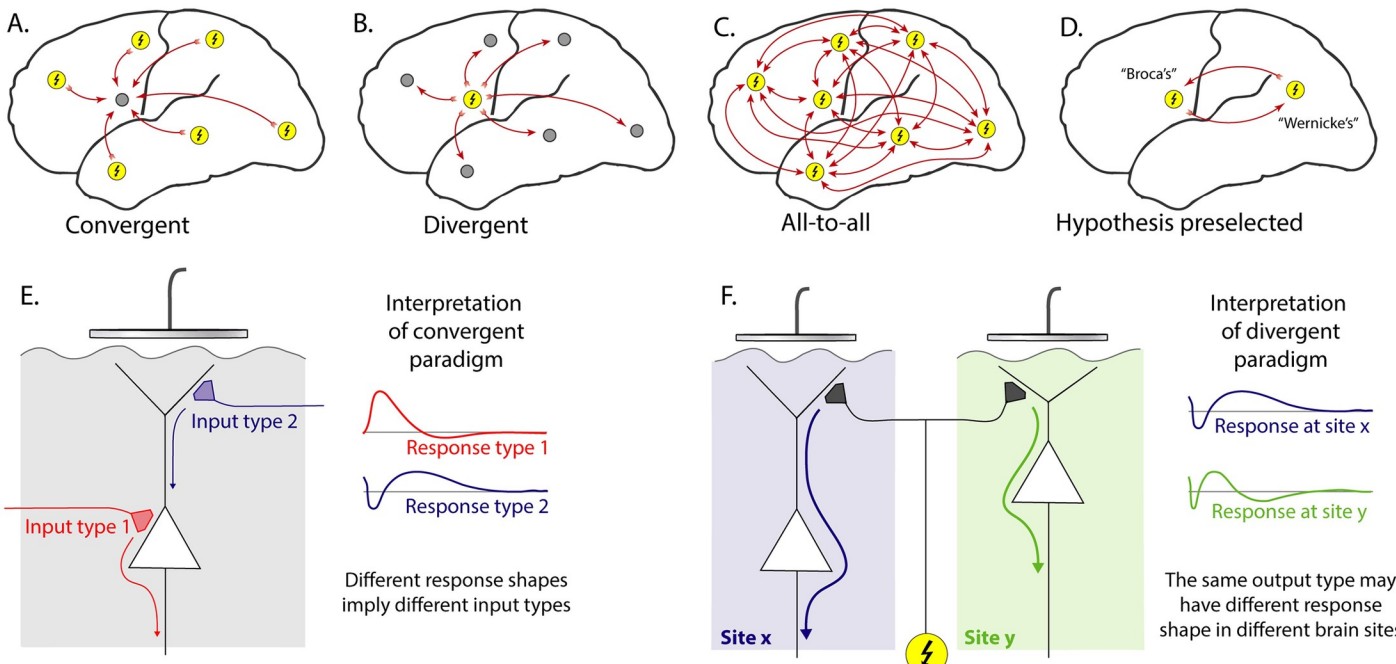

**Fig 1. Cortico-cortical evoked potential analysis paradigms. A**: *Convergent*—Evoked responses at one chosen site (gray circle) are compared with the effect of stimulating all other sites (yellow circles with lightning bolt). For *N* electrodes, this characterizes *N* interactions. **B**: *Divergent*—The temporal response of all sites are examined and compared in response to stimulation of a chosen site (*N* interactions). **C**: *All-to-all*—All *$N^2$* interactions between sites are characterized. **D**: *Hypothesis preselected*—Two sites are chosen based upon a pre-defined anatomical or functional hypothesis, and a 1-way or 2-way interaction between them is characterized. **E**: In the convergent paradigm, all measured responses from a brain surface electrode are associated with the same underlying laminar architecture, so each response shape measured implies a distinct type of input. **F**: In the divergent paradigm, different shaped responses may be measured from different sites, in response to stimulation at a single site. This creates ambiguity because different shaped responses cannot distinguish between 1) the same type of output arriving at cortical sites with different underlying laminar architecture and 2) different types of inputs to sites with similar laminar architecture.

with a compelling spatial clustering [9]. This finding was again observed in the present illustrative dataset, but for responses in the parahippocampal gyrus (PHG) of a different patient, and are shown in Fig 2. While anatomical clustering of stimulation sites that produce similar voltage responses can be observed anecdotally by visual inspection, there is no generally-available quantitative tool for organizing these physiological measurements [7]. Furthermore, we could not find an applicable technique from a different setting that could be translated for the present brain data. Our work aims to close this gap and develop a novel tool to uncover and cluster these temporal motifs in CCEPs and enable systematic exploration of connectivity. We name the resulting canonical voltage response shapes "basis profile curves" (BPC) and detail a framework to identify them.

Within the convergent CCEP paradigm, several criteria constrain the BPC framework. First, there ought to be no assumption of the form of BPCs—they should emerge from the data naturally. Second, they should be able to be mapped into the original data, and onto the brain anatomy in an intuitive way. Third, each stimulation response trial should be able to be parameterized by a single BPC (rather than a superposition of BPCs). Fourth, there should be no orthogonality constraint in BPC shapes in case features (such as the N1 or N2) are shared between BPCs.

Technically, this framework amounts to a hierarchical clustering problem, where the subgroup of single-stimulation events from each stimulation site are known, but how the stimulation site subgroups cluster into the larger group of characteristic stimulation response shapes

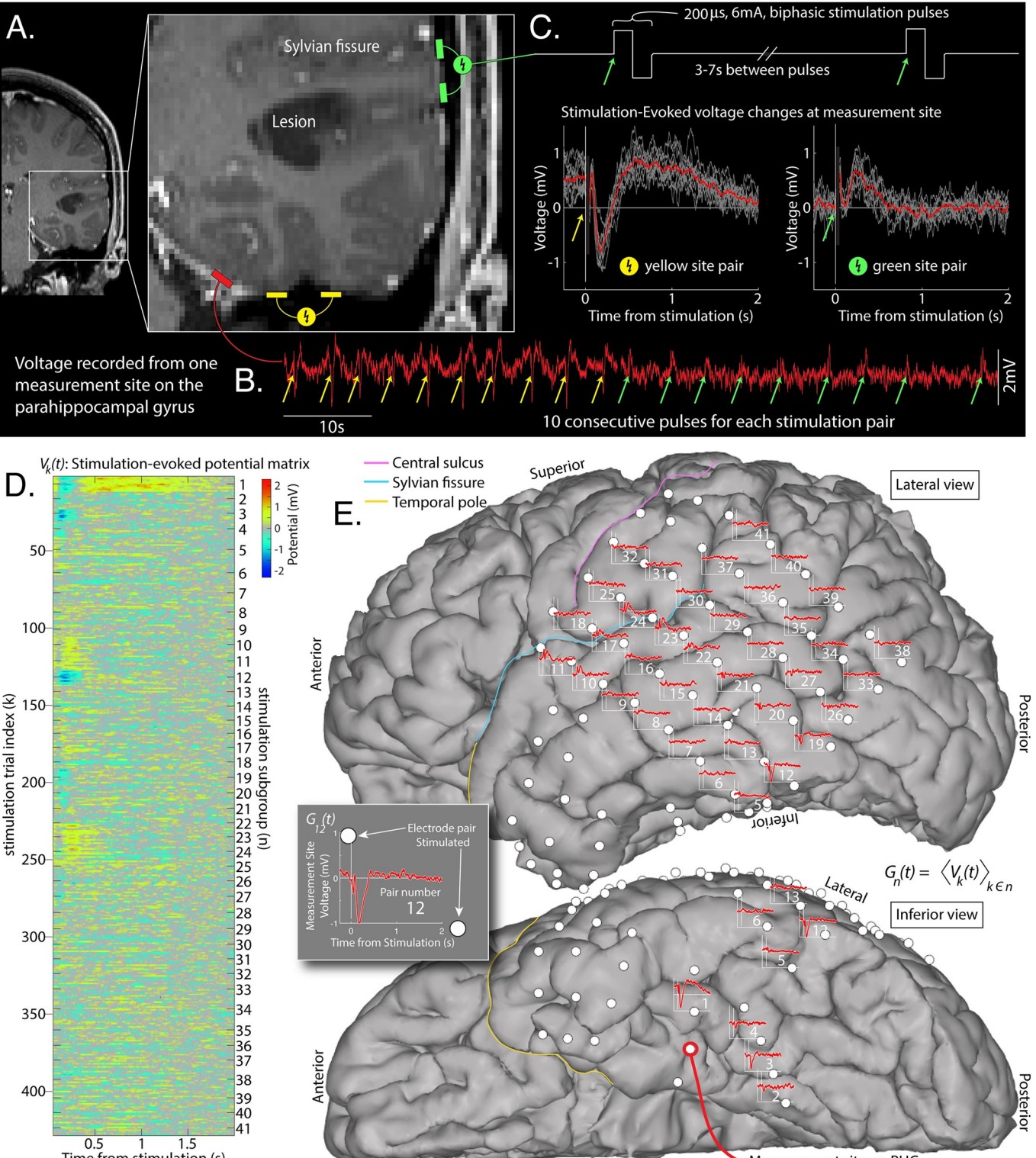

**Fig 2. Single-pulse cortico-cortical evoked potentials. A**: An array of brain surface (ECoG) electrodes were surgically placed on the left hemisphere of a brain tumor patient. **B**: The voltage (red trace) was measured at a parahippocampal gyrus (PHG) electrode site. **C**: Biphasic stimulation pulses were delivered between adjacent electrodes throughout the array (gray shows all stimulation pulse trials for stimulation at each site, red shows average). **D**: Responses from each stimulation pulse are aligned into a matrix $V_k(t)$. **E**: Averaged subgroup responses $G_n(t)$ (i.e. CCEPs, from the PHG measurement site) are shown between the two electrode sites that were stimulated to produce them.

is to be discovered. In this manuscript, we describe a generic solution to this problem, and showcase its potential to obtain novel insights for a representative dataset from implanted ECoG electrodes in a human patient (Fig 2).

## Materials and methods

### Ethics statement

This research was performed in accordance with the Mayo Clinic Institutional Review Board, under IRB# 15–006530, which also authorizes sharing of the data. The patient / representative voluntarily provided independent written informed consent to participate in this study.

### Clinical measurement of cortico-cortico evoked potentials

The patient shown for the illustrative example of this technique was a patient with a left temporal-occipital-parietal tumor (discovered to be a dysembryplastic neuroepithelial tumor) who underwent placement of an electrocorticographic (ECoG) electrode array to localize their seizures and map brain function (Fig 2). This array consisted of a 6x8 frontal-temporal-parietal grid, a 2x8 anterior temporal grid, and 3 1x4 sub-temporal strips of platinum electrodes (Ad-Tech, Racine, WI). The circular electrode contacts had 4 mm diameter (2.3 mm exposed), 1 cm inter-electrode distance, and were embedded in silastic. These arrays were surgically placed on the sub-dural brain surface during staged surgical treatment for functional mapping and seizure localization prior to tumor resection.

Voltage data $V(t_0)$ were recorded at 2048Hz on a Natus Quantum amplifier. Electrode pairs were stimulated $\sim$ 10–12 times with a single biphasic pulse of 200 microseconds duration and 6mA amplitude every 3–7 seconds using a Nicolet Cortical Stimulator (Fig 2A–2C). Electrodes were localized on the CT and coregistered to an MRI using the "CTMR" package [10], available in the "ctmr" folder of the ECoG library [11] or on github [12].

### Data structure (Fig 2)

Data are first structured in a stimulation-evoked voltage matrix $\mathbf{V}$: The time-by-1 matrix $V(t_0)$ for the whole experiment, from the chosen electrode, was sorted into the matrix $V_k(t)$, where $t$ denotes the time from the $k^{th}$ electrical stimulation, $\tau_k$: $(\tau_k + t_1) \leq t \leq (\tau_k + t_2)$. The dimensions of $\mathbf{V}$ are $T \times K$, with $T$ total timepoints (over the interval $t_1 \leq t \leq t_2$) by $K$ total stimulation events (trials). For this illustration, $t_1$ was set to 0.050 sec and $t_2$ to 2.000 sec, so as to begin after the majority of the volume conducted & artifactual changes had passed [13], and to extend until most responses had returned to baseline.

Within the matrix $V_k(t)$, similar events were given a common subgroup label $n$: In the example provided here, each subgroup corresponds to a pair of adjacent electrodes that are stimulated between. The number of repeats is not the same in each electrode stimulation-pair subgroup (typically 10–12). There are $N$ total of these stimulation-pair subgroups. Although we assume that stimulations within each subgroup are independent for the purpose of these analyses, this is an approximation. An example of potential non-independence may be seen in the non-zero offset of individual trials from the yellow site pair in Fig 2C, presumably due to direct charging of the cortical lamina from the prior stimulation pulse due to proximity of the stimulation-pair sites to the recording site.

Trials from the same pair of stimulation sites, $n$, can be combined to obtain the subgroup-averaged evoked voltage change matrix $\mathbf{G}$: For stimulation subgroup $n$, the average voltage temporal profile is $G_n(t) = \langle V_k(t) \rangle_{k \in n}$. With this type of brain stimulation data, $G_n(t)$ are commonly given the name "Cortico-cortical evoked potentials" (CCEPs).

### Single-trial cross-projections & significance matrix (Fig 3A–3D)

In order to understand shared structure between stimulation trials, we first obtain a matrix of unit-normalized single trials: $\tilde{V}_k(t) = V_k(t)/|V_k(t)|$. Each $\tilde{V}_k(t)$ is then projected into all other trials, $\mathbf{P} = \tilde{\mathbf{V}}^\top \mathbf{V}$:

$$P(k, l) = \sum_t \tilde{V}_k(t) V_l(t)$$

Note that $P(k, l) \neq P(l, k)$. The full matrix $\mathbf{P}$ is subsequently sorted into an array of sets $S_{n,m}$ that characterize each cross-subgroup interaction (e.g. pairwise interaction between subgroups $n$ and $m$), such that:

$$S_{n,m} = P\{k \in n, l \in m\}$$

with $k = l$ omitted.

We then construct a matrix of t-values whose elements $\Xi(n, m)$ are the t-values of the corresponding distribution in $S_{n,m}$ (each set's mean divided by its standard error [14]). This matrix is then subjected to a non-negativity constraint, setting negative t-values to zero (a property that is needed for subsequent factorization): if $\Xi(n, m) < 0$, then $\Xi(n, m) \mapsto 0$. $\Xi$ is then scaled to its maximum $\Xi \mapsto \Xi/ \max(\Xi)$, making $0 \leq \Xi(n, m) \leq 1$. This significance matrix plays a role analogous to a cross-correlation matrix to help understand preserved structure of individual trials within, and between different, stimulation-pair subgroups.

Note that the diagonal elements of $\Xi$ can be very small. Interestingly, in some cases, off-diagonal elements are larger than corresponding diagonal element from the same row (Fig 3D). This occurs when there is reliable structure in the response, but, first, within-group variation is larger than the cross-group variation, and, second, there is reliable across-group temporal information across groups (for example, as result of sequence). A simple example of this is when two subgroups produce nearly identical response shapes, but there is adaptation in the response to repeated stimulation for each electrode-pair subgroup—a finding illustrated in some superior temporal gyrus sites from our example (visually apparent in subgroups 11&23 of Fig 3A–3C, and the green in Fig 4C).

### Non-negative matrix factorization (NNMF) for clustering (Fig 3E)

When clustering stimulation sites that produce similar measured responses, a non-negative projection weight constraint must be applied. This is done because the problem is not a source-localization—the same cluster cannot have both positive and negative contributions to the recording site (i.e. trial-to-trial inverted overall sign). Physiologically, this follows because laminar anatomy is not invertible, and when there is positive-negative flip in voltage in one electrode, even if shape is similar, that points to a different biology we want separately segregated into a different cluster (as illustrated in Fig 1E).

The process of non-negative matrix factorization is therefore applied to the matrix $\Xi$, performing a decomposition [15]:

$$\mathbf{\Xi} \sim \mathbf{W}\ \mathbf{H}$$

Where $\Xi$ has dimensions $N \times M$, $\mathbf{W}$ has dimensions $N \times Q$, and $\mathbf{H}$ has dimensions $Q \times M$. The goal of NNMF in this context is to minimize $\eta = |\Xi - \mathbf{WH}|^2$, with the non-negativity constraint $W_{nq}, H_{qm} \geq 0$.

**NNMF multiplicative update rules.**   Begin with randomly generated $\mathbf{W}$ and $\mathbf{H}$, with elements between 0 and 1. The elements of $\mathbf{W}$ and $\mathbf{H}$ are then iteratively updated to better

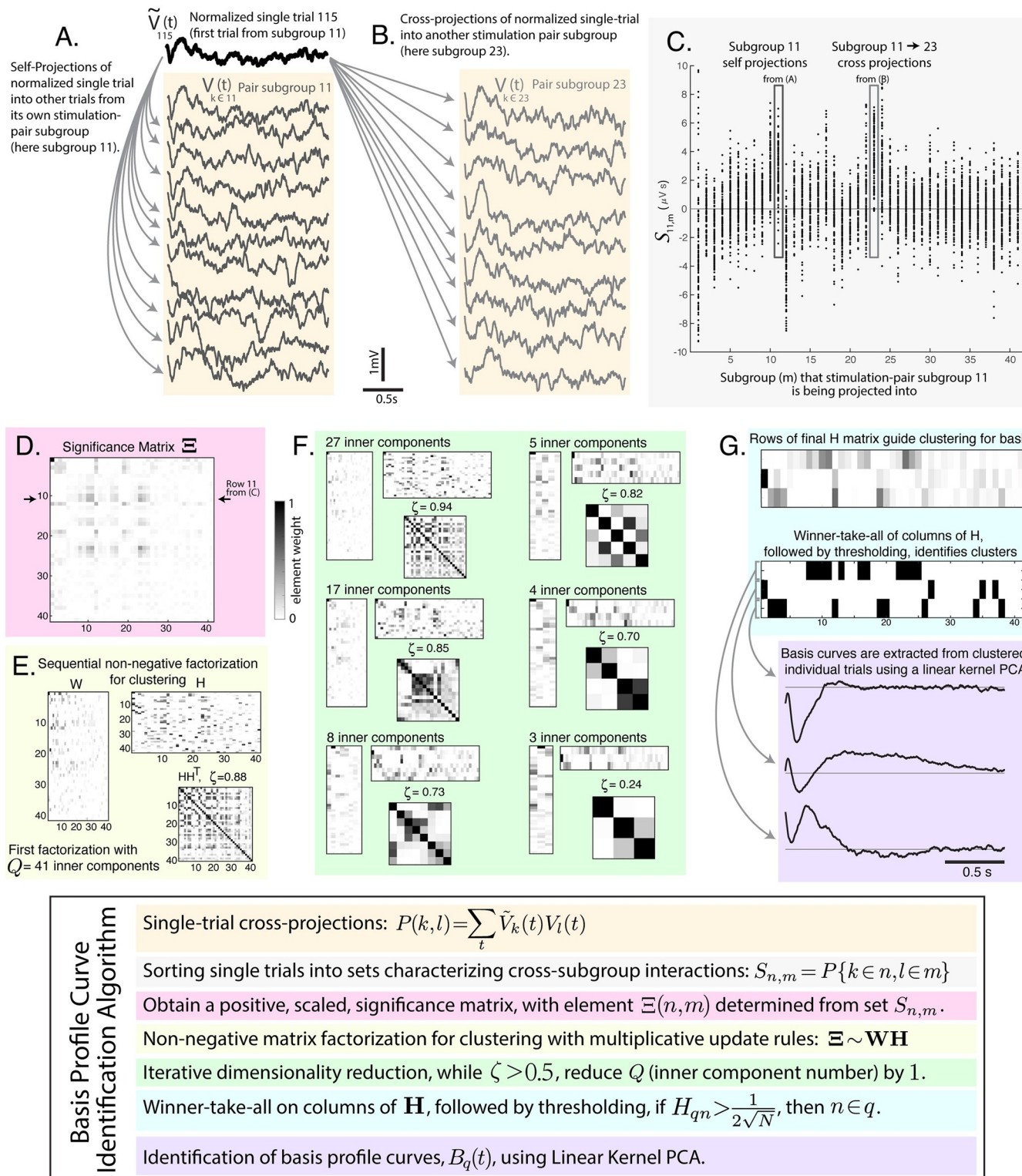

**Fig 3. Technique for identification of basis profile curves (BPCs).** An illustration of the series of steps to extract BPCs from the voltage matrix **V** and subgroup assignments $k \in n$. **A**: Within stimulation-pair subgroup self-projections (all trials are projected into one another). **B**: Between-subgroup cross-projections. **C**: An illustration of sets of cross- and self-projections for stimulation-pair subgroup 11, $S_{11,m}$. **D**: The significance of each set $S_{n,m}$ is determined initially by t-value vs. zero. Negative t-values are set to zero. The matrix of these values is then scaled to 1, and labeled $\Xi$. **E**: Non-negative matrix factorization (NNMF) is performed to identify structure. **F**: The inner dimension of NNMF is iteratively reduced. **G**: BPCs are identified from the groups clustered in the rows of **H**.

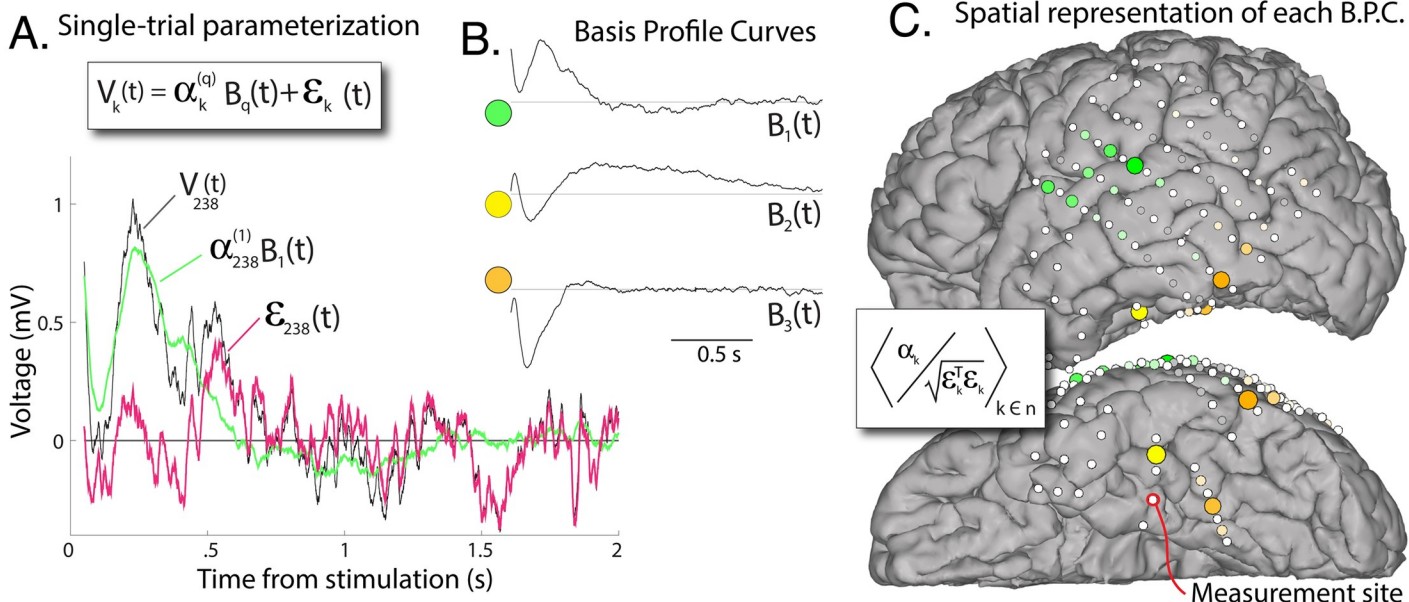

**Fig 4. Projection of basis profile curves (BPCs). A**: The contribution of each BPC $B_q$ to a single trial from its cluster can be quantified according to a scalar multiplier $\alpha_k^{(q)}$, and residual noise $\varepsilon_k$ (trial 238 illustrated). **B**: The 3 BPCs for our example case. **C**: The spatial representation of BPCs, color-coded, with diameter and color intensity indicating magnitude (group-averaged signal-to-noise ratio). White circles show actual electrode locations and BPC projection magnitudes are placed at the the spatial average of the positions of the two stimulated electrodes. Each BPC distribution is individually scaled to maximum. Gray indicates sites discarded by thresholding (Fig 3G).

approximate $\Xi$ until convergence criteria are met. Using a multiplicative (rather than additive) update rule preserves non-negativity in matrix weights [16].

1. First, elements of **H** are individually scaled according to:

$$H_{qm} \mapsto H_{qm} \frac{(\mathbf{W}^\top \boldsymbol{\Xi})_{qm}}{(\mathbf{W}^\top \mathbf{W}\mathbf{H})_{qm}}$$

2. Second, the rows of **H** are unit normalized, maintaining the overall scale by multiplying the columns of **W** by the the normalization factor:

$$H(q, M) \mapsto H(q, M)/|H(q, M)|, \text{ and}$$
$$W(N, q) \mapsto W(N, q) \cdot |H(q, M)|$$

3. Third, elements of **W** are individually scaled according to:

$$W_{nq} \mapsto W_{nq} \frac{(\boldsymbol{\Xi}\mathbf{H}^\top)_{nq}}{(\mathbf{W}\mathbf{H}\mathbf{H}^\top)_{nq}}$$

4. Fourth, the error is calculated as $\eta = |\Xi - \mathbf{W}\mathbf{H}|^2$ and is assessed for convergence, exiting the update loop when the ratio of change in error between subsequent steps to the error is below a set threshold: $\Delta\eta/\eta < 10^{-5}$.

One could alternately choose NNMF with sparseness constraints built into the construction of the factorization algorithm [17], though we defer this to future study. As the process begins

with randomly generated initial **W** and **H**, we re-run the NNMF algorithm a number of times to identify a reliable minimal error $\eta$ between $\Xi$ and **WH**. Separately, one might perform an algorithmic minimization with convergence rather than brute-force repetition, though we found this impractical for two reasons. First, calculation times for re-running the algorithm were quite short in our studies (<3 minutes), and second, approaches to re-parameterizing **W** and **H** for iterative convergence would require significant methodological treatment of their own.

**Dimensionality reduction (Fig 3F).** The output of NNMF can be highly degenerate, and this degeneracy can be quantified by the off-diagonal elements of the matrix $\mathbf{HH}^{\top}$. We define the maximum of the upper-half off-diagonal elements as $\zeta$ (Fig 2F). Then, we iteratively reduce the number of inner components $Q$ by 1 and re-perform the NNMF until $\zeta < 0.5$. This iterative reduction performs our clustering, where the non-zero elements in each 1-by-$N$ row of $H$ define how stimulation-pair subgroups are clustered together. The goal of pruning is to constrain the amount of shared structure in any pairwise comparison of different response shape motifs, and therefore lowering the convergence threshold for $\zeta$ can increase the number of discovered BPCs. We might instead have chosen to constrain the global amount of shared structure by quantifying the sum of the off-diagonal elements of $\mathbf{HH}^{\top}$ (rather than maximum element magnitude).

While we have chosen to start with the beginning number of potential clusters (internal dimension $Q$ in NNMF) to be the same as the number of stimulation-pairs for this example, it is more expedient to start with a lower number of clusters ($\sim 10$ appears appropriate). This is sensible, since the number of basic motifs in laminar organization (at each brain site) might be constrained by a limited number of cell types. One could alternately scan through many different inner dimensions, and select the best in terms of explained variance (see S1 Fig).

**Clustering of subgroups (Fig 3G).** At the conclusion of this process, we perform a "winner-take-all" operation on the columns of **H** such that one stimulation-pair subgroup can only belong to one component (all but the largest elements of each column are set to zero).

For each row $q$ of **H**, a set of stimulation-pair subgroups is assigned to each cluster by thresholding: If $H_{qn} > \frac{1}{2\sqrt{N}}$, then $n \in q$. This threshold is set because if all subgroups contributed equally to a cluster, then the element weight of each would be $1/\sqrt{N}$ (the tolerance factor of 1/2 allows for variation).

This winner-take-all approach implies a process constructed for canonical responses rather than superposition of contributing motifs. For the present work, this assumption is appropriate, though it may not hold in situations where separate physiologic motifs contribute independently, and each may superimpose in the measured response.

Furthermore, in the present example, there is not a significant penalty for inclusion of a group since the significance of this inclusion is later reflected by the scoring of single events, as described below in the subsection "Projecting basis profile curves back into data".

## Identification of basis profile curves using Linear Kernel PCA (Fig 3G)

Parsing the set of all single-stimulation responses from groups that are clustered together by the NNMF process, we can identify characteristic "Basis Profile Curve" (BPC) shapes, $B_q(t)$, in the following manner:

The subset of single trial clustered responses are first concatenated as $V_{k \in n \in q}(t) \equiv \mathbf{V}_{(q)}$, to reflect single trials belonging to stimulation-pair subgroups ($k \in n$) that in turn belong to cluster $q$ ($n \in q$), with a total of $K_q$ such trials. We would like to identify a representative basis curve, $B_q(t)$, that represents the "principal direction" of $V_{k \in q}(t)$. However, the practical fact that the number of timepoints, $T$, generally far exceeds the number of trials, $K$, in these data ($T$

$\gg K_q$) prohibits a standard principal component decomposition (PCA, [18]), which would require $K_q > T^2$ to characterize the $T$-by-$T$ matrix of interdependencies between timepoints.

We address this issue by inverting the decomposition using the "Linear Kernel PCA" technique [19–21]. This method allows for the interchange of an eigenvalue decomposition of the matrix $\mathbf{V}_{(q)}\mathbf{V}_{(q)}^{\top}$ ($T^2$ elements) with $\mathbf{V}_{(q)}^{\top}\mathbf{V}_{(q)}$ ($K_q^2$ elements). Following this approach, we obtain a matrix $\mathbf{F}$, whose columns are the eigenvectors of $\mathbf{V}_{(q)}^{\top}\mathbf{V}_{(q)}$, with associated eigenvalues contained in the diagonal matrix $\boldsymbol{\xi}^2$, satisfying $(\mathbf{V}_{(q)}^{\top}\mathbf{V}_{(q)})\mathbf{F} = \mathbf{F}\boldsymbol{\xi}^2$. We can then solve for the eigenvectors of $\mathbf{V}_{(q)}\mathbf{V}_{(q)}^{\top}$, contained in the columns of $\mathbf{X}$:

$$\mathbf{X}\boldsymbol{\xi} = \mathbf{V}_{(q)}\mathbf{F}^{\top}$$

We keep the first column of $\mathbf{X}$ as our basis curve $B_q(t)$.

Note that, if we were to take the simple average of all candidate stimulation-pair subgroups instead of the 1st principal component for this canonical shape, it could significantly dilute the BPC form with noise. As illustrated in cluster 3 of our example, several low-relevance stimulation-pair subgroups (i.e. very low average projection weight) are clustered along with a strong and significant response subgroup. As we have constructed it, the kernel PCA approach captures the shape of this robust subgroup, without disruption from the uncorrelated noise that dominates the low-relevance subgroups.

## Projecting basis profile curves back into data

Utilizing the formalism from functional data analysis [22, 23], we can represent each individual trial as a projection of a basis profile curve $B_q(t)$, scaled by a scalar $\alpha_k^{(q)}$, with residual error $\varepsilon_k(t)$:

$$V_k(t) = \alpha_k^{(q)}B_q(t) + \varepsilon_k(t)$$

We expect that $E(\varepsilon) = 0$ and $E(\varepsilon_k^2) \sim E(\varepsilon_l^2)$, for all $k$ and $l$. This allows us to estimate the projection of $B_q(t)$ into each individual trial as follows. First, we expand our single-trial formalism above by application of $\sum_t B_q(t)$ to both sides, i.e.:

$$\sum_t B_q(t)V_k(t) = \sum_t B_q(t)\alpha_k^{(q)}B_q(t) + \sum_t B_q(t)\varepsilon_k(t)$$

However, $\sum_t B_q(t)\varepsilon_k(t) = 0$ since $E(\varepsilon) = 0$, and $\sum_t B_q(t)\alpha_k^{(q)}B_q(t) = \alpha_k^{(q)}\sum_t B_q(t)B_q(t)$, which is just $\alpha_k^{(q)}$, since $\sum_t B_q(t)B_q(t) = 1$. This allows us to calculate $\alpha_k^{(q)}$ for each trial:

$$\alpha_k^{(q)} = \sum_t B_q(t)V_k(t)$$

Having $\alpha_k^{(q)}$ determined, we can quantify the residual noise after regressing out the shape of $B_q(t)$:

$$\varepsilon_k(t) = V_k(t) - \alpha_k^{(q)}B_q(t)$$

With the description $V_k(t) = \alpha_k^{(q)}B_q(t) + \varepsilon_k(t)$, several useful quantities for each trial $V_k$ can be described (omitting $q$ for notational simplicity): a "projection weight" $\alpha_k$; a scalar "noise" summary term $\sqrt{\varepsilon_k^{\top}\varepsilon_k}$; a "signal-to-noise" $\alpha_k/\sqrt{\varepsilon_k^{\top}\varepsilon_k}$; the "explained variance" by application of the BPC is $1 - \frac{\varepsilon_k^{\top}\varepsilon_k}{V_k^{\top}V_k}$ (S2 Fig). For stimulation-pair subgroup $n$, we can estimate

the presence of residual structure after application of $B_q$, by the remaining pair-wise correlation in noise terms: $\langle \boldsymbol{\varepsilon}_k^\top \boldsymbol{\varepsilon}_l \rangle_{k,l \in n; k \neq l}$.

## Code and data availability

Code written in MATLAB to reproduce all of the steps and illustrations contained in this manuscript is freely available along with the sample dataset at https://purl.stanford.edu/rc201dv0636. The file "kjm_bpcmethod_readme.pdf" describes the code and dataset, along with instructions for how to perform the analyses.

## Results

Cortico-cortical and subcortical inputs converge in each brain region, and disentangling this convergence will shed light on the networks that interconnect the brain. In order to explore this idea, we stimulated pairs of intracranial electrodes implanted across many different brain areas and measured voltage responses in a single site. We expect that biologically different types of inputs will produce different characteristic shapes in the voltage timecourse, and that these might be clustered into distinct groups based on the stimulation site.

Our framework, aiming to better understand brain connectivity, is grounded in a *convergent paradigm*, examining a set of temporal voltage responses to stimulation, all measured from the same site (Figs 1 and 2). Each response event is labeled by the site of stimulation. Then, a novel algorithm is applied within this framework to identify canonical temporal response motifs, which we call "basis profile curves" (BPCs). Each BPC clusters subgroups of stimulation-pairs together into a larger group whose members induce a similar response profile, and are likely the engaging same microcircuitry in their connectivity from the stimulated brain site to the measured brain site.

### Multiple different stimulation evoked voltage response motifs are measured from one brain site

The BPC approach allowed us to extract a concise set of Basis Profile Curves that describes the multitude of responses observed at a single site. As illustrated in Fig 3, the algorithmic approach begins by obtaining a set of all projection magnitudes (correlations in response timecourse) between pairs of single trials within their own electrode stimulation-pair subgroups and across different stimulation-pair subgroups. A matrix characterizing the significance of each set of subgroup-subgroup projection magnitudes is generated from the t-values of these sets, before setting negative values to 0 and scaling to 1. Then non-negative matrix factorization (NNMF) is repeatedly performed to decompose this significance matrix into a pair of other matrices, one of which characterizes correlated features within the matrix, and the other of which characterizes the weight of each feature (and is normalized). NNMF is performed many times, iteratively reducing the inner dimension of factorization until a cross-correlation threshold between features is surpassed. The iterative reduction clusters electrode stimulation-pair subgroups by their relative element sizes within the rows of the NNMF weight matrix **H** (S3 Fig shows that the technique identifies the optimal number of clusters from the data for portion of variance explained in our example case). For each cluster, linear kernel PCA is applied to the concatenated larger group of single responses from all included stimulation-pair subgroups, to extract the temporal shape of the BPC. In this way, a unique BPC is associated with each cluster. The BPC technique produces an intuitive representation of the responses measured at a single site in the convergent paradigm (S3 Fig). It is robust: Splitting the data in

half yields two similar sets of BPCs, even though there are only 4–5 trials per stimulation-pair subgroup (S4 Fig).

The algorithm, by construction, produces BPCs that have a set of desired properties (Fig 4): Each BPC has a characteristic shape in time of "canonical" responses with simple visual formulation where connectivity between areas is paired with a temporal BPC motif that can be a window into the nature of the interaction. Although the literature has predominantly supported a canonical form for the timecourse of CCEPs (e.g. polyphasic with 2 characteristic "N1/N2" negative deflections), this has not been the case in our measurements. We found instead that the measured cortico-cortical evoked potentials could be described by three basis profile curves, which are unique in shape (Fig 4B), only one of which ($B_3$) is consistent with the reported N1/N2 form.

### Stimulation of adjacent anatomical sites produces similar voltage responses that are clustered together

The initial step in examining these stimulation data from the *convergent paradigm* is to plot the evoked response at the measurement site onto the brain surface at the site of stimulation (Fig 2). Visual inspection of these plots suggests that the response shapes cluster along anatomical boundaries.

Once BPCs have been identified, single stimulation events can be characterized with the architecture from functional data analysis [22–24], which enables quantification of BPC projection weight (signal) and residual noise. These quantifications may be projected back to brain anatomy in an intuitive way to visualize what sites are meaningfully connected to the measured-from site. In our example case (Fig 4), the back projections reveal a relatively sparse network interacting with the PHG originating from the superior temporal gyrus (STG, green in Fig 4), the posterior portion of the inferior temporal gyrus (orange), and the fusiform gyrus (yellow). However, rather than identifying this clustering by visual inspection, the clustering is quantitative with summary weight and clear statistical description. The result is very similar whether quantified by projection weight, signal-to-noise, or explained variance (S2 Fig). Note that the most superior sites included in the STG-BPC (green in Fig 4) may not necessarily result from projections above the Sylvian fissure (e.g. motor areas) to the PHG, but likely result from one of the two electrodes in the included stimulation pair lying on or below the Sylvian fissure.

If a different measurement site is selected in the example patient data, at the temporal pole (TP), a very similar clustered region from the STG emerges (S5 Fig). Interestingly, the shape of the associated BPC is very different. This is precisely the dilemma envisioned by the *divergent paradigm* shown in Fig 1F. The interpretation of the difference in these shapes is ambiguous. One cannot tease out whether the difference in BPC shape implies a different kind of connectivity, or simply a reflection of the different microcircuitry of the TP and STG in response to a similar type of input.

## Discussion

This work begins with the general question of how electrical stimulation paired with voltage measurement can be used to understand spatial and temporal structure in brain networks. Our framework begins with a *convergent approach*, where one measurement site is selected, and responses to sets of repeated stimulations in many other sites are quantified (Fig 1). Each electrical stimulation pulse is brief (<1ms) and the time between pulses is long (>3s), allowing for transient voltage changes to return to baseline. A relatively long time between pulses means that mono-synaptic and polysynaptic effects can contribute to the temporal structure of

the responses, and be captured during data analyses. This experimental paradigm where enough time has passed between consecutive stimulations for transient effects to die out is typically called "single pulse electrical stimulation" [25], a subset of "cortico-cortical evoked potential" (CCEP) measurement. In contrast to the frequently-reported N1/N2 response, our initial observations of these types of data within the convergent paradigm did not suggest a universal form to evoked voltage changes from the same brain site [9]. Therefore, the framework we have developed does not assume what shape the measured responses should have. We instead constructed a data-driven technique to naively extract motifs from all responses at a single seed site that we call "basis profile curves" (BPCs), beginning with a natural set of subgroups defined by repeated stimulations at the same brain site.

## The convergent approach to connectivity

A common approach to explore network structure in CCEP research is to examine every possible interaction within an array of electrodes. In this *all-to-all* approach, the large number ($N^2$) of all possible interactions between the $N$ intracranial electrodes are studied (Fig 1), and each response profile has hundreds-to-thousands of timepoints. The several seconds between each electrical stimulation and the large set of electrode pairs to stimulate between severely limits the number of repeated stimulations that may feasibly be performed in the clinical setting. From a practical perspective, this means that if one does not approach the study of CCEPs with clear constraints, then the problem becomes too high-dimensional to handle, given the limited amount of data available. Many existing studies have attempted to address this by presuming a specific temporal structure in the evoked response, such as the voltage at a pre-specified timepoint post-stimulation [26]. However, as illustrated in Fig 2 and S3 Fig and reported by Kundu and colleagues [27], there is no single canonical CCEP response shape or feature, even when measuring from a single electrode.

Our approach to this high-dimensional data dilemma is to first constrain our study to measurements from a single electrode at a time, leveraging the convergent paradigm. The BPC framework then enables one to naively extract a family of response shapes specific to that site. Although there is no prior assumption about the forms the BPCs should have, they are constructed with the constraint that they should be reproducible within subgroups of repeated stimulations at the same brain site.

Aside from computational convenience, the convergent paradigm is also useful because we can reduce the plethora of potential interactions to a smaller, more tractable, set that may be linked to physiological interpretation (i.e. a few different appearing motifs in interaction between the stimulated and measured sites). For example, inputs to superficial or deeper layers should produce different response motifs in the electric potential measured at the brain surface, and be isolated as distinct BPCs (Fig 1).

Although a single site is selected for the convergent approach, one may iteratively uncover the larger connectivity space: Projections between brain regions can be identified, beginning with a known seed site and then using the proposed algorithm to find a strongly projecting site within a BPC cluster which, in turn, becomes the new seed. This would allow one to trace projections in reverse to explore and model a network of previously unknown interactions.

## Properties of the BPC algorithm

The BPC algorithmic framework is constructed to satisfy a pre-defined set of desired properties, each of which is a associated with distinct computations.

*Simple assignment of each subgroup to a single BPC*: We wanted to meaningfully decompose brain responses to stimulation with a process that can be naturally mapped back on to the

underlying anatomy. This means that stimulated electrode pairs that evoke the same response motif should be grouped together in a way that can be plainly viewed on a brain rendering. In our BPC responses, this means that positive voltage deflections of a particular shape will not be clustered with negative deflections of a similar shape (and vice-versa). Ensuring this is very important when interpreting the signals physiologically. For example, sign flips could reflect inputs at superficial vs deep lamina or at different classes of synapses, and so should be clustered independently [28]. Mathematically, this decomposition can be formulated as a clustering problem with preservation of sub-group structure and inclusion of a non-negativity constraint. Commonly applied techniques such as independent/principal component analyses (ICA/PCA) may not be productive in this setting (even with non-negativity constraints [29]). They assume linear superposition of motifs with arbitrary sign and weight rather than the identification of distinct, unique motifs. In addition, ICA/PCA have different loss functions than the BPC framework, emphasizing different decomposition targets and are unable to reveal the subgroup structure of stimulation-pair sites necessary for our analysis. Specifically, they are specialized to generate components with minimum independency, rather than to cluster motifs by similarity (which is our goal). Canonical correlation analysis (CCA) or variants thereof [30, 31], which might allow for some labeling subgroup structure, do not easily allow one to incorporate necessary constraints and so are of limited help for our purpose.

*Allowance of limited overlap in BPC shape*: Although two different responses may reflect different types of inputs, a limited sub-interval of time may have transient similarity. Therefore, it can be useful to allow for some limited shared structure in the timecourses of different BPCs. In our example, this is illustrated in the negative deflections seen in the initial $\sim$ 500ms of $B_2$ and $B_3$, Fig 4B. We implement this allowance of limited overlap by using a winner-take-all approach rather than enforcing orthogonality in the rows of $\mathbf{H}$ (e.g. $\mathbf{HH}^\top = \mathbf{I}$). The amount of overlap allowed can be adapted by setting the maximum value of individual off-diagonal elements of $\mathbf{HH}^\top$ ($\zeta$).

*Disregarding of meaningless subgroups*: Another physiologically meaningful constraint for the BPC framework is that existing sub-group structure, where responses to stimulation come from the same pair of electrodes should either be reliable on a trial-by-trial basis or not contribute to the clustering. In the BPC approach, a correlation-significance matrix quantifying similarity in single-trial pairwise correlations organized between sub-groups (stimulation-pair sites) can be obtained. Unlike many common decomposition techniques which construct covariance or correlation matrices (ICA/PCA/CCA), the significance matrix generated by this process can have very small diagonal elements, and, in some cases, off-diagonal elements are larger than corresponding diagonal elements from the same row (Fig 3D, see description in Materials and methods subsection "Single-trial cross-projections & Significance matrix"). These small diagonal elements, coupled with the thresholding of the elements of factor matrix *H* excludes stimulation-pair subgroups that do not produce reliable responses in measurement from the identification of BPCs. Once BPCs have been extracted, more explicit quantification of significance for each stimulation-pair subgroup may be evaluated simply, by testing the magnitudes $\alpha_{k,k\in n}$ (for subgroup *n*) versus zero.

*Simple metric to describe single trials*: At the beginning of our data exploration, each single trial *k* is described by its timecourse $V_k(t)$, and the stimulation-pair, $k \in n$, that produced it (its subgroup). At the conclusion of the BPC extraction, each trial is assigned to a single BPC (*q*) with a scalar projection weight ($\alpha_k^{(q)}$), and a residual noise timecourse ($\varepsilon_k(t)$). The parameterization of single-trial stimulation responses takes the form $V_k(t) = \alpha_k^{(q)} B_q(t) + \varepsilon_k(t)$, which is a formalism borrowed from the field of functional data analysis [24]. One might use the tools developed in that discipline for a larger exploration of these data when generalizing across

different patients, tasks, stimulation paradigms, and recording settings. This formalism allows for straightforward characterization of signal and uncorrelated noise in each stimulation single-trial, with comparisons of these across stimulation-pair subgroups and also across BPCs. As noted in the Materials and methods subsection "Projecting basis profile curves back into data", one may quantify a scalar values of signal, noise, and variance explained by the form of the BPC for each single-trial response. These quantities may be useful to plot versus one another (S2 Fig), and allow one to easily scale distributions independently or globally (as seen in S5(D) and S5(E) Fig). The advantage of this formalism moves well beyond that of notational convenience. Namely, the magnitudes of response to stimulation in different brain areas can be directly compared versus one another to quantify how much the local microcircuitry is being influenced, even though the shape of the response timeseries may be very different. Such quantification is typically difficult for timeseries analysis in these types of data, where template-projection or other similar approaches are otherwise necessary [32].

## Response motifs and underlying physiology

At the macroscale that is measured by ECoG or SEEG electrode contact size, an entire population of about half of a million neurons is being averaged over [11]. As such, we expect that only a few BPCs will explain variance from a measurement site. Each region is defined by a unique laminar architecture (defined histologically as a unique Brodmann area [33], or by multi-modal imaging as a unique parcel [34]), so we expect that the pro-dromic (reflecting inputs) or the anti-dromic (reflecting outputs) will be constrained to a few unique motifs, and reflected by relatively few BPCs. Therefore our BPC technique allows for visual inspection of temporal motifs that are well-defined in measurement space (see Fig 4) but raises important questions about the neurophysiology they reflect. Potentially typical shapes may reveal connection to microcircuitry in intuitive way, where surface positive deflections may be tied to deep positive ion influx or superficial ion efflux from specific synapse types in the large pyramidal cells beneath [28]. Alternately, they may reveal projections from the stimulation site directly to different classes of cells within the laminar architecture of cortex (e.g. interneurons vs pyramidal neurons). Some BPC morphologies might reveal different motifs in connectivity at the macroscale, between different brain regions. One might speculate about different types of connectivity within the temporal lobe that may be inferred from examination of Fig 4 and S5 Fig: The preliminary finding of similar spatial distribution in a BPC from the STG, but reflected with a different temporal shape in the TP than the PHG suggests different kinds of projections emanating from a well-defined STG region. For example, direct connections from intracortical axonal projects within gray matter (via lateral projections) might be differentiated from those relayed subcortically through white matter tracts. Note that indirect projections relayed through a third cortical site or a set of subcortical nuclei might each be revealed in characteristic BPC shapes, and this possibility will be explored in future studies.

## Potential future applications

The key elements of our approach are to generate a significance matrix that characterizes group-group similarity, to reduce this matrix by factorization (& thereby perform clustering), and then to recapture the structure that explained this similarity. This framework is novel because it clusters group-labelled measurements into meta-groups, discovering a hierarchical structure in the data. We have explicitly streamlined the framework in order to make the principles clear, and this process has been robust in all of the data we have analyzed. However, future opportunities may also be explored in other settings, adding complexity at each step or introducing bootstrapping/resampling procedural steps [35, 36].

Some examples of what might be attempted are as follows: We implemented linear kernel PCA to identify the BPC underlying a clustered set of trials—this is the simplest strategy other than the simplest average trace, which can add a lot of noise from weakly contributing subgroups. One could instead implement a dictionary learning approach and use the atom that explains the most variance of the included trials [37]. The sequential approach that we use for methodological clarity might instead be expanded into a recursive approach. In a recursive setting, discovered BPCs would be used to re-parameterize the initial trials, these parameterizations would then be clustered, and a new set of BPCs would be found, iterating the process until convergence criteria are met. Alternate approaches to achieve a similar hierarchical clustering with known subgroup labels might begin with a common technique applied to all trials (e.g. k-means, Gaussean mixture model, etc), and then associate these clusters with stimulation-pair subgroups. We have deferred presentation and elaboration of these types of alternate analyses because each would involve a study of the same scope as the present one and distract from the key core elements of the BPC framework.

We believe that initial explorations with our BPC technique would be best framed in a well-controlled setting with connection to simple functional studies from primary motor, auditory, or visual regions to look for commonality in BPC shapes. In this manner, existing understanding from other types of measurement might serve to validate BPC interpretation. From our initial studies, natural branching out from the single seed site of the convergent paradigm will lead researchers to examine the generalization of BPC shapes discovered at adjacent measurement sites within a brain region, and may also lead them to explore whether they are conserved at homologous brain sites across different patients. Such "second-level" studies of BPC shape across contexts would employ more traditional clustering approaches than the present strategy. For example, measurements of the brain's depths with stereoelectroencephalography should allow for sign flips when comparing BPCs from adjacent electrodes spanning superficial vs deep cortical layers.

Future work will also examine whether specific BPC shapes are associated with similar biological motifs convergent on different brain regions (e.g. thalamocortical relays, U-fiber projections, or a common interneuron-projection neuron structure). While this work does not explicitly address biomarkers of disease state, one might hypothesize that seizure networks and onset zones will have altered dynamics reflecting epileptogenicity, with corresponding abnormality in BPC shape.

Subsequent studies might vary experimental conditions with a task, medication, or property of the stimulation (current magnitude and temporal profile, or timing between pulses), and quantify signal, noise, and residual structure in the responses. For example, one could examine the effect of focusing on a pair of brain locations (i.e. in the *hypothesis preselected* paradigm) and simply changing the amplitude of stimulation. It has been demonstrated that different stimulation magnitudes can elicit very different morphologies [27], and one could study this by labeling each stimulation amplitude as a different subgroup, and then examining the distribution of BPCs that emerge.

Inspecting recordings from a single site, our novel framework could be applied to behavioral, rather than electrical stimulation. For example, one could study higher order visual areas, like the fusiform gyrus, and examine responses to presented images of different semantic types [38, 39]. In such a study, one would replace stimulation pair groups with semantic stimulus groups (pictures of faces, houses, tools, etc.) to see how different semantic groups cluster together in production of the fusiform electrophysiological response.

This "winner-take-all" approach (Fig 3G) detailed in this BPC framework implies a process constructed for canonical responses rather than superposition of contributing motifs. In other words, each stimulation response is assigned to a single BPC, rather than as a combination of

multiple BPCs. For the present work, this assumption is appropriate, though it may not hold in situations where separate physiologies contribute independently and may superimpose in the measured response. One example of this would be where evoked responses to electrical stimulation may vary based upon changes in region-specific ongoing fluctuations in neuronal excitability. If threshold levels must be met before specific components of a composite response are elicited (e.g. a delayed second voltage deflection), then this method (which forces a unitized shape) would be sub-optimal, and another strategy should be employed. There may be future approaches where initial voltage changes (1$^{\text{st}}$ order projections) and later (2$^{\text{nd}}$ & higher order projections) are assessed independently and allowed to superimpose, particularly when attempting to detect/discover potential higher-order responses. Such a strategy could be performed by picking multiple sets of $t_1$ and $t_2$, and re-performing analyses for isolated time windows.

Lastly, a feature of this functional data analysis formalism which might be explored in future work is that candidate structures not correlated to initial subgroup labeling could be tested for by looking at correlations of $\langle \varepsilon_k^\top \varepsilon_l \rangle_{k \neq l}$, where $k$ and $l$ would be of a candidate different type of subgroup (for example, the third stimulation pulse from each set of stimulations). In this case, after a full exploration of that residual structure, the error term $\varepsilon$ for a given trial would split out into a second order set of BPCs, $C_p$, with coefficients $\beta_k^{(p)}$, where $\varepsilon \rightarrow \beta^{(p)} C_p + \varepsilon'$, and $V_k(t) = \alpha_k^{(q)} B_q(t) + \beta_k^{(p)} C_p(t) + \varepsilon'_k(t)$.

## Conclusion

We have detailed a new data-driven framework for uncovering motifs in epoch-based time-series data belonging to labeled subgroups. These motifs are called "basis profile curves" (BPCs), and they determine characteristic spatiotemporal structure. Each timeseries epoch is assigned to a unique BPC, with motif projection strength and residual noise simply parameterized. This framework is applied to understand the effect of electrical stimulation in the human brain using arrays of implanted electrodes, where the labeled subgroups are repeated stimulations at the same site. We introduce a set of paradigms for interpreting these measurements, of which the *convergent* one allows application of the BPC framework. In our illustrative example of measurements from the surface of the parahippocampal gyrus, we find that identified BPCs clearly uncover several connected regions and allow them to be viewed and interpreted intuitively.

## Supporting information

**S1 Fig. Explained variance as a function of number of inner dimensions, $Q$.** We can quantify the total variance explained by the fitting of resulting BPCs, $B_q(t)$, across all included stimulation-pair trials: $\langle V_k^\top V_k - \varepsilon_k^\top \varepsilon_k \rangle_{k \in n \in q}$. When we normalize this by the total variance, $\langle V_k^\top V_k \rangle_{k \in \text{ all } n}$, we have the portion of total variance explained (blue circles). However, the goal of the decomposition is to identify motifs, explained by BPCs, from the full set of stimulations. For example, if one were to perform stimulations at a site that has no effect on the measurement site, it should not undermine our confidence in the decomposition. Therefore, a more appropriate normalization is to instead divide the explained variance by the variance of the trials included in the clustering, $\langle V_k^\top V_k \rangle_{k \in n \in q}$. For our example case, this immediately validates the dimensionality $Q = 3$, selected by the algorithm (Fig 3).
(TIF)

**S2 Fig. Alternate metrics for projection weights of BPC curves.** As in Fig 4, but using alternate scoring metrics for each stimulation-pair subgroup, shown in inset rectangles overlying

each cortical rendering.
(TIF)

**S3 Fig. Side-by-side illustration of CCEP shapes (from Fig 2E) and BPC projection weights (from Fig 4B and 4C), to plainly illustrate the correspondence.**
(TIF)

**S4 Fig. An illustration of the BPC method applied to sub-segmentation of trials.** On the top row, BPCs are identified from only odd trials. On the bottom row, BPCs are identified from only even trials. After the split, each stimulation-pair subgroup is represented by 4–5 individual stimulation trials. Despite this small amount of individual stimulation trials for each stimulation-pair subgroup, the process is remarkably stable. For each row, the panels from left-to-right are matrices of the CCEPs, extracted BPC shapes, projection weights (group-averaged signal-to-noise ratio), and projection weights plotted on the brain surface.
(TIF)

**S5 Fig. Illustration from a different site.** Stimulation responses from a site in the temporal pole are shown. **A**: Responses from each stimulation pulse are aligned into a matrix $V_k(t)$. **B**: Averaged responses $G_n(t)$ are shown at the site of each stimulation pair that produced them. **C**: BPCs produced by the algorithm. **D**: Weights (group-averaged signal-to-noise ratio) associated with each BPC (color-coded), and non-included sites (gray). **E**: Spatial representation of BPCs, color-coded, with diameter and color intensity indicating magnitude. All values are scaled to the global maximum across all BPCs. **F**: As in (E), but with each BPC distribution individually scaled to its own maximum. Note the similarity in the spatial distribution of the stimulation-site cluster labeled in green to Fig 3), but the completely different shape of the BPC (likely reflecting the different laminar architecture of the two recipient measurement sites).
(TIF)

## Acknowledgments

We are grateful for the participation of the patient in the study, and for the assistance of the staff in Saint Marys Hospital at Mayo Clinic, Rochester, MN. Greg Worrell, Pradeep Udayavar Shenoy, Amin Nourmohammadi, and Harvey Huang generously provided helpful discussion and commentary during the drafting of the manuscript.

## Author Contributions

**Conceptualization:** Kai J. Miller, Klaus-Robert Müller.

**Data curation:** Kai J. Miller, Dora Hermes.

**Formal analysis:** Kai J. Miller.

**Funding acquisition:** Kai J. Miller, Klaus-Robert Müller, Dora Hermes.

**Investigation:** Kai J. Miller, Klaus-Robert Müller, Dora Hermes.

**Methodology:** Kai J. Miller, Klaus-Robert Müller.

**Resources:** Kai J. Miller.

**Software:** Kai J. Miller.

**Validation:** Kai J. Miller, Klaus-Robert Müller, Dora Hermes.

**Visualization:** Kai J. Miller, Dora Hermes.

**Writing – original draft:** Kai J. Miller.

**Writing – review & editing:** Kai J. Miller, Klaus-Robert Müller, Dora Hermes.

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
