## [Decision Letter · Decision Letter 0]

25 Mar 2021

Dear Dr. Miller,

Thank you very much for submitting your manuscript "Basis profile curve identification to understand electrical stimulation effects in human brain networks" for consideration at PLOS Computational Biology.

As with all papers reviewed by the journal, your manuscript was reviewed by members of the editorial board and by several independent reviewers. In light of the reviews (below this email), we would like to invite the resubmission of a significantly-revised version that takes into account the reviewers' comments.

Three reviewers have considered your paper and they were all enthusiastic about the innovative methods you developed. However, all had concerns that need to be addressed before publication. The main concern, in my opinion is that this method needs to be compared-in theory and in practice-to existing methods for solving this problem.

I encourage you to respond to these comments and submit a revised version of the manuscript.

We cannot make any decision about publication until we have seen the revised manuscript and your response to the reviewers' comments. Your revised manuscript is also likely to be sent to reviewers for further evaluation.

Sincerely,

Marieke Karlijn van Vugt, PhD

Associate Editor

PLOS Computational Biology

Daniele Marinazzo

Deputy Editor

PLOS Computational Biology

Three reviewers have considered your paper and they were all enthusiastic about the innovative methods you developed. However, all had concerns that need to be addressed before publication. The main concern, in my opinion is that this method needs to be compared-in theory and in practice-to existing methods for solving this problem.

I encourage you to respond to these comments and submit a revised version of the manuscript.

Reviewer's Responses to Questions

**Comments to the Authors:**

Reviewer #1: The authors describe a novel technique for understanding brain connectivity using the CCEP paradigm. The development of novel methods for using brain stimulation to map connectivity is an important question across a number of clinical and cognitive neuroscience domains. Overall, I think the method proposed by the authors represents a valuable addition to the literature surrounding this issue. In particular the use of a single analytical pipeline incorporating all observed stimulation sites is an important innovation.

The method involves identifying similarity in the response curves to stimulation to understand how a number of spatially distinct anatomical stimulation sites may be connected to a single site of interest. The multi-step approach includes generating correlation matrices for 1) a single stim site to itself across stim trials and 2) other stimulation sites, extracting similarity features, and then projecting these onto the observed responses to generate intuitively useful response curves for the different locations

The principal question I have is related to how this technique performs relative to straightforward existing methods, such as comparing evoked responses. Do the BCP derived curves shown in Fig 3 and then again in Fig 4 look like ERPs from the different stimulation sites? Could the same discernments in connectivity motifs arise from existing methods, such as examining N1 and N2 components or measuring correlations in the ERPs for example? More generally, other updated methods have been proposed for measuring CCEP responses (https://doi.org/10.1016/j.jneumeth.2019.108559, https://doi.org/10.1016/j.neuroimage.2017.07.061). The paper would benefit from engaging with basic, traditional methods and some of these updated methods and an explanation of why this newer method is more useful. One of the key differences in this method is that the authors do not average response to stimulation at a single site but examine response curves for all trials, which introduces some issues with shifts in baseline (as they acknowledge). They should make this clear when comparing their technique to others. As a point of clarification, is every stimulation trial mapped to every other stimulation trial in the initial step of the pipeline? Or just the first trial? This was not entirely clear to me from the figure.

The authors propose some uses of CCEP mapping but seem to ignore the principal application, which is understanding seizure networks (https://doi.org/10.1016/j.eplepsyres.2015.04.009). Readers considering application of the technique would be interested to know if it may have utility in this area. Related to this point, one of the issues with CCEP has been how to control for the effect of expected response magnitude due to anatomical distance (including volume conduction https://doi.org/10.1016/j.jneumeth.2020.108639). How does this new method handle this issue? Based on the data in Fig 4, it seems that significant response motifs are either nearby (green) or further away (yellow), but most sites near the electrode of interest do not show a significant BPC response. Are strong responses not observed for nearby locations because the resulting responses are not consistent and end up being removed as noise? Or were they just not stimulated? Additional plots showing how the new method handles this issue would be a nice addition to the manuscript.

Related to this, additional explanation would be helpful in Fig 4. Most important is to show clearly where the measured response is located on the brain, the site to which the colored circles are presumably connected (what is the “convergent” electrode site?). This is more clear in the supplemental figure for temporal pole.

The authors include a paragraph describing the additional studies needed to verify the utility of this new method. I understand that this manuscript focuses on the methodological details. But for a high profile journal, I think a demonstration that this method identifies a connection of known anatomical effective connectivity (such as the arcuate fasiculus, visual-fusiform connections, as published for early validation of basic CCEP methods) would make the argument more convincing. Comparing how the BPC method performs compared to existing methods for such a known connection would be helpful.

How do the authors handle multiple stimulation amplitudes? They report a single 6 mA threshold, but because of differences in impedance across stimulating locations (due to heterogeneity in electrode contact with the cortex), application of CCEP usually includes several stimulation thresholds (eg 2 to 8 mA). Why did the authors use only a single amplitude in their input data? They mention this as a future application, but I would assume several stimulation amplitudes are available for the example they show based on how CCEP data are collected. Do different stim amplitudes at the same region identify significant motifs (perhaps with different shapes after projection onto the observed response curves)?

Finally, I don’t know if this method would be useful in stereo EEG (depth electrode) data. The variation in the positive/negative deflection attributed to different interactions of depth electrodes with gray matter led to the adoption of techniques such as root mean square or gamma power to measure CCEP responses (https://doi.org/10.1016/j.cortex.2019.05.019). This same issue impacts the use of evoked potentials using stereo EEG recording electrodes. Based on the description provided in page 13, my sense is that the authors would say that this method is useful only for surface electrodes, where the relationship to the cortical surface is consistent, but perhaps this is incorrect. Given the explanation of how their method maps onto the neurophysiological properties of the brain, if the authors believe it would be useful in stereo EEG, showing some data to this effect would be important.

Typos I noticed: Line 226

Line 263

Reviewer #2: In this manuscript, Miller et al. outline a conceptual and computational approach to “convergent” brain stimulation mapping. They argue that, by stimulating many different sites and measuring the potentials evoked at a single target site (convergent approach), we generate a more interpretable dataset than by stimulating one site and measuring the resulting potentials at many other recipient sites (divergent approach). They further provide a computational approach for convergent mapping, based on a two stage algorithm: non-negative matrix factorization followed by linear kernel PCA. Their methods are illustrated with an example dataset and perform well in this setting.

Overall, this is an important, insightful and useful contribution to the theoretical framework around electrical brain stimulation. The main areas for improvement in the manuscript relate to the textual justification of the two-stage electrode-assignment and waveform-assignment approach, and clarification of some of the writing and presentation around the algorithm.

MAIN POINTS

1) The computational scheme outlined here seeks to satisfy two goals: (G1) identify groups of locations whose stimulation elicits similar waveforms at the target site, and (G2) to extract canonical waveforms (basis profile curves, BPCs) and associate them with individual stimulation events. There are conceptual and practical questions about this consecutive two stage approach.

Conceptually, it seems important to state explicitly what is the relative priority of each of these goals. In other words, suppose it were possible to achieve a better or more ambiguous solution to (G1) while rendering (G2) less interpretable or less accurate — which of these goals takes priority? You could imagine this as a kind of objective function in which we are optimizing for both goals — what is the weighting on each term in the objective function?

Practically, I could not help wondering whether a recursive procedure might not work better to jointly achieve the two goals (G1) and (G2). In many clustering settings [see for example, k-means clustering, but the principle applies more generally to all kinds of expectation-maximization algorithms] a broad approach is to (Step 1) make a preliminary clustering of trials and then (Step 2) extract canonical waveforms form each cluster, and then return to step (Step 1) using the canonical waveforms extracted in (Step 2) as a reference or basis of some kind. Conversely, in the current paper the two stages (clustering trials / electrodes, and then extracting BPCs) are performed strictly consecutively. Could the BPCs not be used as a means of performing better clustering? Do the authors think that the consecutive approach taken here is better than a recursive methods, or simply more straightforward?

2) A key assumption of the decomposition (lines 63-65) is that each stimulation trial should be characterized by a single BPC rather than a superposition of BPCs. I would like to hear more about whether this is a computationally expedient assumption, or whether it is endorsing an assumption about the nature of the mapping process and brain dynamics. In particular, it seems that if there are ongoing fluctuations in the excitability of the neurons in the target site, then this alone could give rise to variation in the trial-to-trial mixture of responses elicited in a single site. For example, consider the case where we are always stimulating exactly the same adjacent electrode-pair and recording from the same target site. Now, suppose that one BPC is associated with “high pre-stim excitability of pyramidal neurons at the target site” and another is associated with “low pre-stim excitability fo pyramidal neurons at the target sites”… would one not expect that a superposition of these BPCs would be observed on stimulation trials on which the pre-stimulation excitability in the target site is “intermediate excitability”?

3) At a broader level, I would have liked to have seen a few sentences in the Discussion addressing the broad question: “How would we know if the results of this method are better than another method? What should we treat as ground truth, or the practical outcome which can be used to decide that one decomposition approach is ideal”? I could imagine pure neuroscience arguments here (e.g. consistency in extracted clusterings when using convergent mapping for target site A and target site B) and more applied outcomes (e.g. better performance in identifying pathological brain tissue). Ultimately, though, I think that a few sentences on this topic could help to remind the reader what all of these methods are for, and to provide a framework for comparing these methods against any other approaches.

4) For selecting the BPCs (paragraphs around lines 162 and 180) , is the selection based on explained variance ideal, or could it be improved using a cross-validation based selection procedure? [Hold out some of the data; fit the model; measure the goodness of reconstruction in the held-out data]. The authors need not implement this, but out-of-sample explained variance may be worth mentioning as an alternative to in-sample explained variance.

5) Figure 4: Could the authors comment on why the ventral visual system is so responsive to stimulation in the ventral motor cortex? Obviously resolving this is not important for this manuscript, but it seems worth commenting on the spatial distribution shown in Figure 4C, in light of the very ventral site (PHG) that is being targeted.

6) Line 292: “We found instead that the measured cortico-cortical evoked potentials could be described by three basis profile curves, which are unique in shape (Figure 4B), only one of which (B3) is consistent with the reported N1/N2 form. “This seems like an important point — at the same time, when making an observation like this it seems important to confirm for the reader that the N1/N2 forms are “ostensibly” present in the raw data. Because another reason why they may not be recovered by this procedure could be that they are simply not present in this example dataset… so you would want to show that the N1/N2 components //are// (under some conventional appraisal) present in the raw data (perhaps this is just done by eye?) and yet these N1/N2 peaks are not recovered as BPCs.

7) Although I find the arguments in favor of the “convergent” mapping procedure compelling (as opposed to divergent), I think that what the authors present here — mapping of /two/ sites using convergent mapping and then comparing the maps — is even more compelling, because it enables you to see whether the same site-clusters (e.g. the motor cortex cluster) are identified as clusters for two different target sites. If the authors agree, then they may want to consider moving Supp Figure 3D/E into the main text?

……

MINOR POINTS

0) Throughout the manuscript “figure 2” and “figure 3” references could be “Figure 2” and “Figure 3”.

1) Might sparse dictionary learning methods be used in place of the linear kernel PCA approach? (e.g. Mairal et al., 2009, Online Dictionary Learning for Sparse Coding. ICML).

2) In Figure 2, the site of each stimulation site is mapped on the cortical surface. Is this “site” of stimulation the spatial average of the positions of the two stimulated electrodes?

Line 17: this sentence seems awkward toward the end:

“For an array of N total electrodes, there are a potential set of order N2 CCEP interactions that may be explored (for bipolar stimulation, the actual number will depend on which sets of adjacent stimulation pairs are chosen to deliver electrical pulses through).”

should “stimulation pairs” be “electrode pairs”?

Line 27: “and then study” could be “and then to study”

Line 101: “T total time points” could be “T time points”

Line 103: May want to make this more concrete for the reader by saying something like, “In the example provided here, each subgroup corresponds to a pair of adjacent electrodes”

Line 120: the notation in the equation for S_{nvm} is confusing , with k an element of n, and l an element of m… if n and m are sets, would it not be clearer to demarcate them with capital letters or some other token?

Line 124: this matrix is first subjected to a non-negativity constraints — please clarify if “first” here means before or after t-value conversion… presumably it would have to happen after the t-value conversion? In that case it would not be the first manipulation? Maybe “next” rather than “first”?

Line 132: “there is structure in the variation itself” — maybe this would be clearer as “there is reliable temporal structure in the within-group variation” ..?

Line 156: “pushing the normalization factor onto the columns of W” — please clarify

Line 159: the previous steps were written in passive tense and this is now in an imperative tense (“calculate” vs. “is calculated”)

Line 182: I think this could be “it is more expedient to start with…”

Line 193: please clarify where the tolerance factor was added, and the sensitivity of the procedure to the choice of this parameter

Line 226-7: “several insignificant stimulation-pair subgroups” — please make clear what “insignificant” means in this context — is this a statistical statement or estimate?

Line 236: please check punctuation here … the structure of the sentence is “Since X, and Y=W=Z”

Line 250 : disentangling of this convergence —> disentangling this convergence

Line 255: based upon where was stimulated —> based on the stimulation site

Line 268: a single site —> at a single site

Line 279: “by their relative weights” — please specify which parameter this refers to

Line 358: “Although there is no prior assumption about the forms the BPCs should have, …” — perhaps specify by mentioning e.g., continuity or differentiability or variance or something like that… certainly the decomposition scheme must impose /some/ assumptions and constraints on the form of the BPCs… it is just a fairly generic time-series decomposition assumption… I would imagine that this matrix decomposition technique could be re-written as a kind of Bayesian inference procedure, with some priors over the variables, correct?

Line 446: “Each region is defined by a unique laminar architecture (defined as a unique Brodmann area, [32]),…” — Is this actually true? My understanding was that cytoarchitectonic features would sometimes vary in a graded manner, and that regional boundaries are drawn based on multiple features, including cortical thickness and myelination and other factors? This is a very small point… it just seems strange to cite Korbinian Brodmann for this claim … maybe consider citing a more modern atlas, too, like some of the recent Van Essen group atlases based on multimodal imaging?

Reviewer #3: I would like to commend the authors for their submitted manuscript, in which they clearly outline a novel way to cluster data resulting from medical singe pulse electrical stimulation (SPES) sessions in intracranially implanted patients. Indeed their approach appears to provide a data-driven way to quantitively tease apart different response profiles resulting from stimulation throughout the brain, which they call basis profile curves" (BPCs). This method provides clusters that cannot be obtained to more commonly employed methods (for example ICA, direct PCA based methods etc), which would probably cluster the 3 BPCs categorized in the example into the same cluster (possibly even producing a traditional N1/N2 profile), which would not reflect the single trial-level accurately. This method allows for data resulting from SPES sessions to be analysed in more depth and with higher accuracy, while still constraining the amount of variables. I am enthusiastic about this method and can see many potential applications in future analyses. The manuscript is very well-written and surely will benefit iEEG research.

I have only two concerns, one major and one minor (and list a few typos). I hope these help the authors to improve on the manuscript.

Specific comments

1. Major. The proposed approach will work very nicely when one considers only one electrode where signals are recorded. However, in many cases a researcher will have several electrodes in a given area. For instance, one may have 5 electrodes in the hippocampus and uses SPES data to test how the hippocampus is connected to other brain regions. This problem would even be more excacerbated when one has mircowire recordings, and increasingly popular tool, where there are 8 channels per location. The approach presented by the authors would provide a handful of BPCs per channel, but how can we group these BPCs across electrodes in a meaningful way? Notably, when grouping across electrodes, phase-reversals (ie. polarity flips) are possible (even expected) so the NNMF wouldn’t work in this instance. It would be useful if the authors could adapt their toolbox to implement such a second-level clustering solution. However, the authors may wish to answer that this question is beyond the scope, which is fair enough, but then at least they should offer some guidance as to how BPCs across electrodes can be grouped.

2. Minor. This point relates to the brute-forcing of the non-negative matrix factorization (NNMF). You mention that alternative implementations to repeatedly re-running the NNMF were deemed impractical. I would like to see more detail about why that is the case. Is this due to intense computation requirements/times, that offset the advantage of reducing the randomness of W and H? I was also wondering how much variability this method leaves in resulting BPCs in data that are a bit more noisy/not as clear cut.

Typos:

Line 144: … anatomy IS not …

Line 268: … observed AT a single site …

Line 435: there is a ‘)’ missing in the reference to suppl fig 2.

**Have all data underlying the figures and results presented in the manuscript been provided?**

Reviewer #1: Yes

Reviewer #2: Yes

Reviewer #3: Yes

PLOS authors have the option to publish the peer review history of their article (what does this mean?). If published, this will include your full peer review and any attached files.

Reviewer #1: No

Reviewer #2: No

Reviewer #3: **Yes: **Simon Hanslmayr
---

## [Decision Letter · Decision Letter 1]

21 Jul 2021

Dear Dr. Miller,

We are pleased to inform you that your manuscript 'Basis profile curve identification to understand electrical stimulation effects in human brain networks' has been provisionally accepted for publication in PLOS Computational Biology.

Best regards,

Marieke Karlijn van Vugt, PhD

Associate Editor

PLOS Computational Biology

Daniele Marinazzo

Deputy Editor

PLOS Computational Biology

Two of the reviewers accepted the submitted manuscript without further revisions, and a third reviewer was also mostly satisfied, although this person still had some remaining comments. Nevertheless, since these comments seek to expand the scope of the paper but do not question the validity of the results, I am convinced the paper is ready for acceptance. However, I encourage the authors to take a look at the comments provided by this reviewer, especially their suggestions for changes in wordings. Apart from that, congratulations!

Reviewer's Responses to Questions

**Comments to the Authors:**

Reviewer #1: The authors have modified their manuscript to add explanation (including modification to Fig 2) regarding the relationship between the BPCs and evoked responses observed via stimulation. I think this is a helpful addition, in that it emphasizes the similarity between existing ERP methods and the BPC curves, which will make the new method easier to grasp (at least it did for me). I apologize for my oversight of the “measurement site” label in the figures – reading it properly makes the figure easier to interpret also.

The authors’ point about different response curves at different recordings sites (as summarized in Fig 1F and citation of Kundu) is exactly the motivation behind the adoption of RMS as a method for measuring connectivity using CCEP. I understand the focus of the manuscript is to present the BPC method, but I don’t think that showing how their method contrasts with results from RMS (for the single patient they describe) represents an undue distraction. In fact, the authors might point out that while RMS would perhaps identify both the green and orange dots as highly connected to the PHG, the BPS can differentiate the response patterns. The authors do a good job explaining the limitations of N1/N2 patterns for inferring connectivity, but given the ubiquity of RMS in the analysis of CCEP I think a direct contrast would be helpful. One idea would be to report where in the distribution of RMS values the significant (green and orange) locations fall.

This is also why I don’t entirely understand why the BPC method would be useful in SEEG, because different response shapes will be observed for electrodes located in the same brain region (due to different spatial relationship relative to the cortical surface). If the response curves have a different shape, then they would be separately clustered by the BPC method (I assume) even though they have similar connectivity. I understand the authors want to limit the scope of the study, but I think any strong claim about utility for BPC in SEEG would require some data so they could perhaps change the wording here.

I understand the authors’ point regarding their desire not to add more data to the manuscript examining BPC for regions with known connectivity (eg arcuate fasiculus). I still think, for a high profile journal, this would be a useful addition, especially since the connections shown in Fig 4 are somewhat surprising (peri-sylvian locations connected strongly to the PHG). But if no data are available beyond this individual patient then I can understand omitting such a validation since collecting new data might be infeasible in humans. I would also be surprised that no data are available at lower stimulation amplitudes at least, principally because stimulating at 6 mA at all locations might induce seizures in sensitive locations and usually lower amplitudes need to be tested before moving to higher amplitudes. But I agree that showing how the method performs at different stim amplitudes is not critical to the manuscript.

Reviewer #2: I thank the authors for their revisions, which have addressed all of my concerns.

Reviewer #3: The authors have done a great job in addressing our comments. We have no further suggestions and congratulate the authors on a great paper.

**Have the authors made all data and (if applicable) computational code underlying the findings in their manuscript fully available?**

Reviewer #1: Yes

Reviewer #2: Yes

Reviewer #3: Yes

PLOS authors have the option to publish the peer review history of their article (what does this mean?). If published, this will include your full peer review and any attached files.

Reviewer #1: No

Reviewer #2: No

Reviewer #3: No

---

## [Editor Report · Acceptance letter]

11 Aug 2021

PCOMPBIOL-D-21-00124R1

Basis profile curve identification to understand electrical stimulation effects in human brain networks

Dear Dr Miller,

I am pleased to inform you that your manuscript has been formally accepted for publication in PLOS Computational Biology. Your manuscript is now with our production department and you will be notified of the publication date in due course.

With kind regards,

Andrea Szabo
